# Tau phosphorylation suppresses oxidative stress-induced mitophagy via FKBP8 receptor modulation

**Michael O. Isei[1], Meredith Crockett[1], Emily Chen[1], Joel Rodwell-Bullock[1], Trae Carroll[2], Peter A. Girardi[1], Keith Nehrke[2]\*, Gail V. W. Johnson[1]\***

**1** Department of Anesthesiology & Perioperative Medicine, University of Rochester, Rochester, New York, United States of America, **2** Nephrology Division, Department of Medicine, University of Rochester, Rochester, New York, United States of America

\* gail_johnsonvoll@urmc.rochester.edu (GVWJ); keith_nehrke@urmc.rochester.edu (KN)

**Data Availability Statement:** All relevant data are within the manuscript and its Supporting information files.

## Abstract

Neurodegenerative diseases are often characterized by mitochondrial dysfunction. In Alzheimer's disease, abnormal tau phosphorylation disrupts mitophagy, a quality control process through which damaged organelles are selectively removed from the mitochondrial network. The precise mechanism through which this occurs remains unclear. Previously, we showed that tau which has been mutated at Thr-231 to glutamic acid to mimic an Alzheimer's-relevant phospho-epitope expressed early in disease selectively inhibits oxidative stress-induced mitophagy in *Caenorhabditis elegans*. Here, we use immortalized mouse hippocampal neuronal cell lines to extend that result into mammalian cells. Specifically, we show that phosphomimetic tau at Ser-396/404 (EC) or Thr-231/Ser-235 (EM) partly inhibits mitophagy induction by paraquat, a potent inducer of mitochondrial oxidative stress. Moreover, a combination of immunologic and biochemical approaches demonstrates that the levels of the mitophagy receptor FKBP8, significantly decrease in response to paraquat in cells expressing EC or EM tau mutants, but not in cells expressing wildtype tau. In contrast, paraquat treatment results in a decrease in the levels of the mitophagy receptors FUNDC1 and BNIP3 in the presence of both wildtype tau and the tau mutants. Interestingly, FKBP8 is normally trafficked to the endoplasmic reticulum during oxidative stress induced mitophagy, and our results support a model where this trafficking is impacted by disease-relevant tau, perhaps through a direct interaction. We provide new insights into the molecular mechanisms underlying tau pathology in Alzheimer's disease and highlight FKBP8 receptor as a potential target for mitigating mitochondrial dysfunction in neurodegenerative diseases.

## Introduction

Accumulation of abnormally phosphorylated tau (pTau) and damaged mitochondria are key contributors to neurodegeneration in Alzheimer's disease (AD) [1–4]. While the causative relationship between pTau and mitochondrial dysfunction is still debated, studies in

**Funding:** National Institute on Aging NIH R01 AG067617. The funders had no role in study design, data collection and analysis, decision to publish, or preparation of the manuscript.

**Competing interests:** NO.

transgenic models of tauopathies and human AD patients indicate that pTau disrupts mitochondria quality control by mitophagy, leading to oxidative neuronal damage [1, 5, 6]. Evidence suggests that accumulation of damaged mitochondria due to impaired mitophagy correlates more strongly with cognitive deficits in AD than amyloid burden [3, 7]. Yet, the specific molecular mechanisms whereby pTau disrupts mitophagy remain unclear.

Mitophagy is a cellular housekeeping cascade that involves (i) identification of defective mitochondria, (ii) enveloping them with a phagophore containing microtubule-associated protein-1 light chain-3 (LC3), (iii) targeting the mitochondria-phagophore complex (mitophagosome) to the lysosome, and (iv) degrading the contents of the mitolysosome. The PINK1/parkin and receptor-mediated mitophagy pathways are well characterized [8, 9] and involve proteins that may be targets of pTau toxicity [3, 10]. Extensive research has explored the PINK1/parkin-mediated mitophagy pathway under stressed and basal conditions [10–12]. However, studies on receptor-mediated mitophagy are limited, despite its critical role in maintaining mitochondrial quality under stress [13]. Understanding receptor-mediated mitophagy and how it is influenced by pTau may provide insights for identifying potential therapeutic targets for AD.

Receptor-mediated mitophagy (S1 Fig) involves transmembrane receptors on the outer mitochondrial membrane, with a cytoplasmic LC3 interacting region (LIR) that binds LC3 on the phagophore [13, 14]. Mammalian mitophagy receptors include BCL2/adenovirus E1B 19-kDa-interacting protein 3 (BNIP3), FUN14 domain-containing protein 1 (FUNDC1), NIX, Bcl2-like 13 (BCL2L13), and FK506-binding protein prolyl isomerase 8 (FKBP8/FKBP38) [15, 16]. Generally, mitophagy can be triggered by mitochondrial depolarization, pharmacological agents, hypoxia, and electrolyte loss [9, 14], with reports suggesting receptor selectivity to different stressors [17]. For example, FUNDC1, BNIP3 and NIX are classically induced by hypoxia [13, 18] while FKBP8 is mainly activated by mitochondria-targeted stressors [17].

FKBPs, a family of immunophilins highly expressed in the nervous system particularly in AD susceptible brain regions, are crucial for cell signaling and neuroprotection [19]. Notably, FKBP8, the only FKBP with a transmembrane domain, serves as a mitophagy receptor by preferentially interacting with LC3A to recruit phagosomes to mitochondria [16, 20]. Unlike other mitophagy receptors, FKBP8 mostly avoids lysosomal degradation during mitophagy by translocating from mitochondria to the endoplasmic reticulum (ER) [16, 21, 22]. The impetus for this escape is not fully understood but is likely crucial to sustain mitophagy and inhibit apoptotic signaling during mitophagy [21]. While studies observed decreased expression of non-mitophagy-related FKBPs in AD brains such as FKBP12 and FKBP52 [23, 24], the role of FKBP8 in AD remains unclear, and the impact of pTau on FKBP8-mediated mitophagy is unexplored.

Our focus is on AD-associated tau phosphorylation because its accumulation strongly correlates with AD progression and mitochondrial dysfunction [25, 26]. Thus, we created phosphomimetics of tau at Thr231/ Ser235 (EM) or Ser396/Ser404 (EC). Focusing on double phosphorylation as modeled by EC and EM, rather than the early AD-associated single-site T231E we used in our previous studies in *C. elegans* [27, 28], better mimics the complex phosphorylation patterns of tau observed later in AD [29]. This approach provides a more accurate model for understanding the impact of pTau on mitophagy in a non-aging cell culture model [25, 30, 31]. It is of interest to explore how these phosphomimetics impact mitophagy. Identifying proteins affected by these pTau species during mitophagy could clarify their roles in AD pathogenesis and uncover potential therapeutic targets. Previously, we showed in *C. elegans* that tau T231E selectively inhibited oxidative stress-induced mitophagy [27]. Here, in immortalized mouse neuronal hippocampal cells (HT22 cells), we found that EC and EM phosphomimetics suppress oxidative stress-induced mitophagy and the inhibitory effects may result

from pTau interfering with FKBP8 receptor-mediated mitophagy pathway and dynamics at the mitochondrial-ER interface.

## Materials and methods

### Reagents and antibodies

Potassium chloride (KCl, #22C1556440) and magnesium chloride (MgCl$_2$, #20C1956719) were from VWR life science. Sucrose (#210555) was from Fisher Bioreagent. Ethylene glycol bis (β-aminoethyl ether)-N, N′-tetraacetic acid (EGTA) was from MilliporeSigma. Paraformaldehyde was from Alfa Aesar (#43368). Fetal clone II (FCII, #AF29578030) was from Cytiva Hyclone. Dulbecco's modified Eagle medium (DMEM, #2418283), Neurobasal media (#21103–049) gentamicin (#2328211), and GlutaMax (#2380959) were from GIBCO, and Paraquat (#227320050: Methyl viologen hydrate 98%) was from Thermo Fisher Scientific. Puromycin (#13884) was purchased from Cayman chemical company, while NP-40 (#119929) was obtained from USB Corporation. Antibodies: FKBP8 (Proteintech, 66690-1-Ig), PINK1 (Proteintech, 23274-1-AP), parkin (Proteintech, 14060-1-AP), beta actin (Cell signaling, 8457L), GAPDH (Santa Cruz Biotechnology, sc-25 778), beta tubulin (Proteintech, 10094-1-AP), TOM20 (Protein tech, 11802-1-AP), FUNDC1 (ABclonal, A16318), BNIP3 (Abcam, 431196). The tau antibodies: total tau (Dako tau, A0024), phospho-Ser396/404 (was a gift from Dr. P. Davies), phospho-T231/S235 (Cell signaling, 20473S). HRP-conjugated mouse anti-rabbit IgG (Cell Signaling Technology, 5127S), human tau (5A6, Developmental Studies Hybridoma Bank), total tau Tau-5 (Thermo Fisher Scientific, AHB0042). Anti-mouse IgG HRP-conjugated antibody (Bio-Rad, 5178–2504). Anti-rabbit IgG HRP-conjugated antibody (Bio-Rad 51,962,504). Alexa Fluor™ 488 donkey-anti-mouse (Thermo Fisher Scientific, A32766). Alexa Fluor™ 594 donkey-anti-rabbit (Thermo Fisher Scientific, A-21207).

### Construction of mt-mKeima-P2A-tau

Briefly, PCR fragments for the UBIC promoter, mt-mKeima-P2A fusion, and 0N4R wild-type human tau were amplified from vectors Fig B (a generous gift from Dr. C. Pröschel, University of Rochester), pHAGE-mt-mKeima-P2A-FRB-Fis1 [32], and eGFP-Tau [33], respectively. Amplified fragments were fused together via overlap extension PCR, or "PCR stitching" [34]–restriction sites and the required fragment homology were introduced using the primers below. The UBIC::mt-mKeima::P2A::0N4R Tau cassette was gel purified (Qiagen Cat#28704), restriction digested using SphI and NotI, and ligated into pLVX-EF1α-eGFP-2x-strep-IRES-Puro [35]. The final vector (pTC9) was fully sequenced via Plasmidsaurus. To generate the phosphomimetic tau constructs, where threonine and/or serine were changed to glutamic acid (EC: pS396E/S404E and EM: pT231E/S235E), we PCR amplified EC and EM fragments from our pEGFP-EC and pEGFP-EM vectors, respectively [33]. The mutated-tau fragments were purified, digested with Blp1 and Not1, gel extracted and used to replace the Blp1-NotI fragment in pTC9.

Amplifying UBIC from Fig B:

F: 5'– AGAGCATGCGATGGTTAATTAACCCACCC – 3'

R: 5'– ACATGGTGGCGTCTAACAAAAAAGCCAAAAACG – 3'

Amplifying mt-mkeima-P2A from pHAGE-mt-mKeima-P2A-FRB-Fis1:

F: 5'– TTTGTTAGACGCCACCATGTCCGTCCTGAC – 3'

R: 5'– GCTCAGCCATAGGTCCAGGGTTCTCCTCCA – 3'

Amplifying 0N4R Tau from eGFP-Tau Vector:

F: 5'– CCCTGGACCTATGGCTGAGCCCCGCC – 3'

R: 5'– CTTGCGGCCGCTCACAAACCCTGCTTGGCC – 3'

## Cell culture, DNA transduction, and generation of stable cell line

HT22 cells, originally gifted to us by Dr. P. Hemachandra Reddy from Texas Tech University, Lubbock, TX, were maintained at 37˚C and 5% $CO_2$ in DMEM supplemented with 10% heat-inactivated fetal bovine serum, 2.2 mM GlutaMax, and 25 μg/mL gentamicin. pTC9 and mutated constructs were packaged into lentiviral particles using psPAX2 (Addgene, #12260) and VSV-G (Addgene, #12259) and transduced into sub-confluent HEK293TN cells using PolyJet (SignaGen Laboratories, SL100688). After 12 h, the media was replaced with DMEM containing 1% FCII. Supernatant from the transfected cells was collected after 48 h, filtered using a 0.2 μm syringe filter, concentrated by ultracentrifugation at 30,000 g for 3 h at 4˚C. The concentrated virus was aliquoted, snap frozen, and stored at -80˚C until use. Two days post transduction, puromycin selection was carried out until uninfected cells were lost. Surviving colonies were expanded and maintained in media supplemented with 0.5 μg/mL puromycin. Tau and mt-mKeima proteins expression were assessed by western blotting and fluorescence microscopy, respectively.

## Immunoblotting

Stable cell lines were grown on 10 cm plates to 70% confluence, rinsed with PBS, and harvested into cell lysis buffer (0.5% NP-40, 150 mM NaCl, 10 mM Tris-Cl [pH 7.4], 1 mM EGTA, 1 mM EDTA, 1 mM phenylmethylsulphonyl fluoride, and 10 μg/ml each of aprotinin, leupeptin, and pepstatin). Aliquots of cleared lysate were run on 12–15% SDS-acrylamide gel, then transferred onto a nitrocellulose membrane as we described previously [36]. For antibody incubation, both primary and secondary antibodies were diluted in 5% non-fat milk dissolved in Tris-buffered saline containing 0.05% Tween 20 (TBS-T). Primary antibodies were incubated overnight at 4˚C, followed by a 1 h incubation at room temperature with HRP-conjugated anti-rabbit or anti-mouse secondary antibody. Signals were detected with Immobilon Crescendo Western HRP-substrate (Millipore, WBLUR0500) and Kwik Quant imager (Kindle Biosciences, LLC) and normalized to beta-actin or Ponceau-S staining to adjust for protein loading.

## Mitophagy assessment

Oxidative stress-induced mitophagy was measured using the pH-sensitive fluorescent protein mt-mKeima, a red fluorescent emitter targeted to the mitochondrial matrix that is resistant to acid proteases and has a bimodal excitation [37]. Mitophagy was induced by incubating the cells with 5 μM paraquat (PQ) for 6 h. At neutral pH such as in the mitochondrial matrix the 440 nm excitation predominates, whereas at acidic pH such as in the lysosome the 550 nm excitation predominates [38]. This unique characteristic allows mt-mKeima fluorescent ratiometry to be used as a surrogate to measure mitophagy, or mitochondrial targeting and engulfment by lysosomes. Images were taken under constant acquisition conditions with Zeiss microscope equipped with Colibri 7 LED, Axiocam 705 camera, and custom filter sets from Zeiss. These sets were designed to excite at 440 nm and 550 nm and emit at 620 nm. To significantly minimize photobleaching, the excitation power was reduced to 5%. Additionally, image

acquisition was limited to fields containing one or two cells to enhance analysis accuracy and mitigate potential interference from confluent cells. Image analysis was performed using ImageJ. After subtracting background intensity, images in both channels (440 nm and 550 nm) were split. Mitochondria and mitolysosomes in respective channels were thresholded to generate regions of interest. Subsequently, the analyze and set measurement functions were used to calculate the area and intensity and exported as Microsoft Excel data. Mitophagy index was defined as the ratio of the area and intensity of the mitolysosome signal (550 nm) to that of mitochondrial signal (440 nm).

## Co-immunoprecipitation

Stable cells grown on 10 cm plates were rinsed twice with PBS and collected in immunoprecipitation lysis buffer (150 mM NaCl, 50 mM Tris-HCl, 1 mM EDTA, 1 mM EGTA, and 0.5% NP-40, pH 7.4). The lysates were sonicated for 10 sec at 0.5 (Misonix Inc, S-3000) and then centrifuged for 10 min (16,000 g, 4°C). The resulting supernatants were transferred into 1.5 ml centrifuge tubes, and protein concentration was determined using the bicinchoninic acid (BCA) assay. For co-immunoprecipitation of tau and FKBP8, 500 μg protein was incubated with rabbit IgG control or rabbit anti-tau antibody overnight at 4°C and pulled down by protein A/G Magnetic Agarose beads (Thermo Scientific, 88802) for 6 h at 4°C. The beads were washed three times with wash buffer (2 mM EDTA, 0.1% NP-40) and twice with lysis buffer. Proteins were eluted from the beads using 2 × SDS-PAGE buffer by boiling for 10 min at 95°C. The samples were then resolved by Western blotting, probing the nitrocellulose membrane with mouse anti-FKBP8 antibody.

## Immunocytochemistry

Stable cells lines were cultured on 18mm coverslips coated with poly-d-lysine (PDL; Sigma Aldrich, A003E) in 12-well plates. The cells were fixed in 4% paraformaldehyde-sucrose for 10 min and permeabilized with 0.25% triton x-100 in PBS for 10 min, then blocked with 5% BSA and 300mM glycine in PBS for 1 h. Subsequently, the cells were incubated with FKBP8 (1:1000, anti-rabbit) and BiP/GRP78 (1:1000, anti-mouse) primary antibodies overnight. Appropriate Alexa Fluor™ conjugated secondary antibodies (Fluor™ 488 and Fluor™ 594) were then added at room temperature for 1 h, protected from light. Cells were then washed three times with PBS, and coverslips were mounted on glass microscope slides using Fluoro-Gel Mounting Medium (Electron Microscopy Sciences, 17985–30). Images were taken with Nikon A1R HD scanning confocal microscope using a 40x objective and analyzed using Imaris 10.0 software (Oxford Instruments). The colocalization function was used to analyze twenty cells from 5-10 images in three independent experiments using Mander's colocalization coefficients module to quantify fluorescence overlap.

## Mitochondria isolation

Mitochondria fractions were isolated as described by Wang et al. 2008 [39], with modifications. All samples were maintained on ice and all centrifugations were performed at 4°C. Cells were gently rinsed twice with PBS and scraped in ice-cold mitochondrial isolation buffer (MIB: 250mM sucrose, 10mM Tris-HCl, 10mM KCl, 1mM EGTA, 1.5mM MgCl$_2$, 1mM DTT, 0.1mM PMSF, 1X protease inhibitor cocktail). The cells were homogenized by 30 passages through a 1 ml 26G ½ insulin syringe and centrifuged at 800 g for 15 min. The supernatant was then centrifuged at 14,000 g for 15 min. The supernatant and pellet thereafter are the cytosolic and mitochondrial fractions, respectively. The mitochondrial fraction was washed twice by resuspending in mitochondria respiration buffer (MRB: 250mM sucrose, 10mM Tris-HCl,

10mM KCl, 1mM EGTA, 1.5mM MgCl$_2$, 1mM DTT, 0.1mM PMSF, 1X protease inhibitor, 0.5% NP-40) and centrifuging at 11,000 g for 10 min. Finally, the pellet was resuspended in 100μl MRB and protein concentration was measured by BCA assay.

### FKBP8 knockdown in stable tau cell lines

DNA oligonucleotides with overhangs (Fwd: 5'-CTA GCT CAG CAA GGT GAA TAT AGT ttcaagaga ACT ATA TTC ACC TTG CTG AGC TTT-3'; Rev: 5'-TCG AAA AGC TCA GCA AGG TGA ATA T AGT tctcttgaa ACT ATA TTC ACC TTG CTG AG-3') of mouse FKBP8 mRNA, were designed using BLOCK-iT™ RNA Designer (Thermo Fisher Scientific) and annealed accordingly. The annealed oligonucleotides were inserted into the XbaI and XhoI-digested pHUUG vector (a generous gift of from Dr. C. Pröschel) and the ligated products were amplified in competent cells. Subsequently, the vector was packaged into lentiviral particles using psPAX2 (Addgene, #12260) and VSV-G (Addgene, #12259) and transduced into sub-confluent HEK293TN cells using PolyJet (SignaGen Laboratories, SL100688). After 12 h, the media was replaced with DMEM containing 1% FCII. Supernatant from the transfected cells was collected after 48 h, filtered using a 0.2-μm syringe filter, concentrated by ultracentrifugation at 30,000 g for 3 h at 4°C, and resuspended in Neurobasal media. Cells were transduced for 7 days prior to use. Immunoblotting was used to confirm knockdown.

### Statistics

Statistical analyses and graphs were made using GraphPad Prism software v 9.0. Student's *t*-test was used for pairwise comparisons, while comparisons in multiple groups were analyzed with one-way or two-way analysis of variance (ANOVA), followed by Tukey's *post-hoc* test. Normality of distribution was tested using Kolmogorov-Smirnov test. Significance was denoted as * ($p < 0.05$), ** ($p \leq 0.01$), *** ($p \leq 0.001$), **** ($p \leq 0.0001$). Graphs show individual data points with bars representing mean value, while the error bars represent the standard error of the mean (SEM).

## Results

### Phosphorylated tau suppresses oxidative stress-induced mitophagy

To develop a cell culture model for testing the impact of phosphorylated tau on mitophagy, HT22 cells were transduced with lentiviral particles expressing tau or phosphomimetic tau mutants 2EC (S396E/S404E) or 2EM (T231/S235) as a P2A protein fusion with an N-terminal fluorescent mitophagy reporter mt-mKeima under the control of the UBC promoter. Cells stably expressing the transgene were enriched under puromycin selection. Tau expression was verified via western analysis using antibodies against total tau, pS396/404, and pT231/S235 (Fig 1). Each of the stable lines expressed relatively equivalent amounts of tau, and as expected the antibodies recognizing specific phospho-epitopes did not react with corresponding mutants where Glu was substituted for Thr or Ser (Fig 1). In addition, the stable lines expressed less tau overall than mouse primary neurons (S2 Fig).

Each stable line also expressed mt-mKeima, a mitochondrion targeted fluorescent protein whose excitation maxima differs depending upon the surrounding pH. In the mitochondrial matrix, the pH is neutral and the 440-nm excitation dominates, whereas lysosomes are acidic, and when mitochondria are taken up through mitophagy, the excitation maxima shifts to 550-nm [38]. Mt-mKeima was targeted appropriately to the mitochondria as shown by colocalization with Mitotracker CMX-Ros (S3 Fig or Fig 2A). Although the Mitotracker dye fluoresces in the red spectrum much like mt-mKeima under acidified conditions (in

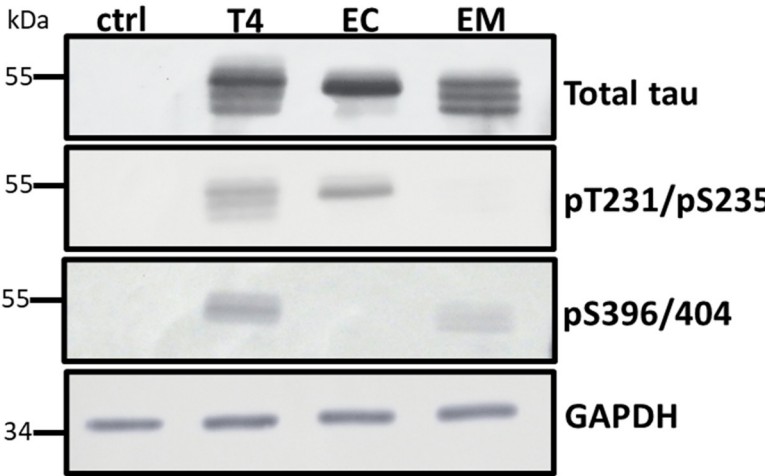

**Fig 1. Tau expression in stable HT22 cell lines.** Cell lysates (20 µg protein) from cells expressing wildtype (T4) and phosphomimetic tau (EC and EM) were analyzed by Western blot using site-specific and phosphorylation-dependent anti-tau antibodies. Tau expression levels were comparable across all groups. The control lane (ctrl) confirms the absence of endogenous tau in HT22 cells.

mitolysosomes), and one might expect overlapping signal, in practical terms the magnitude of the Mitotracker signal dwarfed that of mitolysosomes at baseline (Fig 2A) and spectral separation was possible in mitochondrion due to separate excitation wavelengths (440-nm vs. 550-nm).

To quantify mitophagy, an index was calculated representing the ratio of sum fluorescent intensities of the mitochondrial signal (440-nm ex/620-nm emission) and mitolysosome signal (550-nm ex./620-nm em.). As mentioned, under resting conditions the baseline mitophagy index was very low (Fig 2C). However, when the cells were treated with PQ, a complex I inhibitor and redox cycler that stimulates mitochondrial oxidative stress, mitolysosome abundance–consisting of multiple puncta with clearly distinct morphology from mitochondria themselves—was greatly increased (Fig 2B) and the mitophagy index rose many-fold (Fig 2C). Most importantly, cells expressing wildtype tau exhibited ~2-fold greater increase than cells expressing 2EC or 2EM (Fig 2C). This suggested that the disease-relevant 2EC and 2EM pTau mimetics at least partly suppress oxidative stress-induced mitophagy.

## Mitophagy inhibition may be due to phosphorylated tau's impact on FKBP8 receptor

Recent reviews have proposed that AD-associated pTau influences mitophagy by affecting proteins integral to mitophagy pathways [40]. To elucidate how pTau suppresses mitophagy in our experimental model, we performed immunoblot analysis to assess its impact on the expression of key molecular mediators and mitochondrial proteins involved in both PINK1/parkin- and receptor-mediated mitophagy pathways.

We first examined the expression of PINK1 and parkin, essential regulators of the PINK1/parkin mitophagy pathway [41]. Upon induction of mitophagy with PQ, PINK1 and parkin expression levels remained largely unchanged across all tau groups (Fig 3A–3C). This suggests that PINK1/parkin mitophagy contributes marginally to oxidative stress-induced mitophagy in our model system. Next, we focused on the receptor-mediated mitophagy pathway, assessing the expression of FKBP8, FUNDC1, and BNIP3 receptors. These proteins are known to

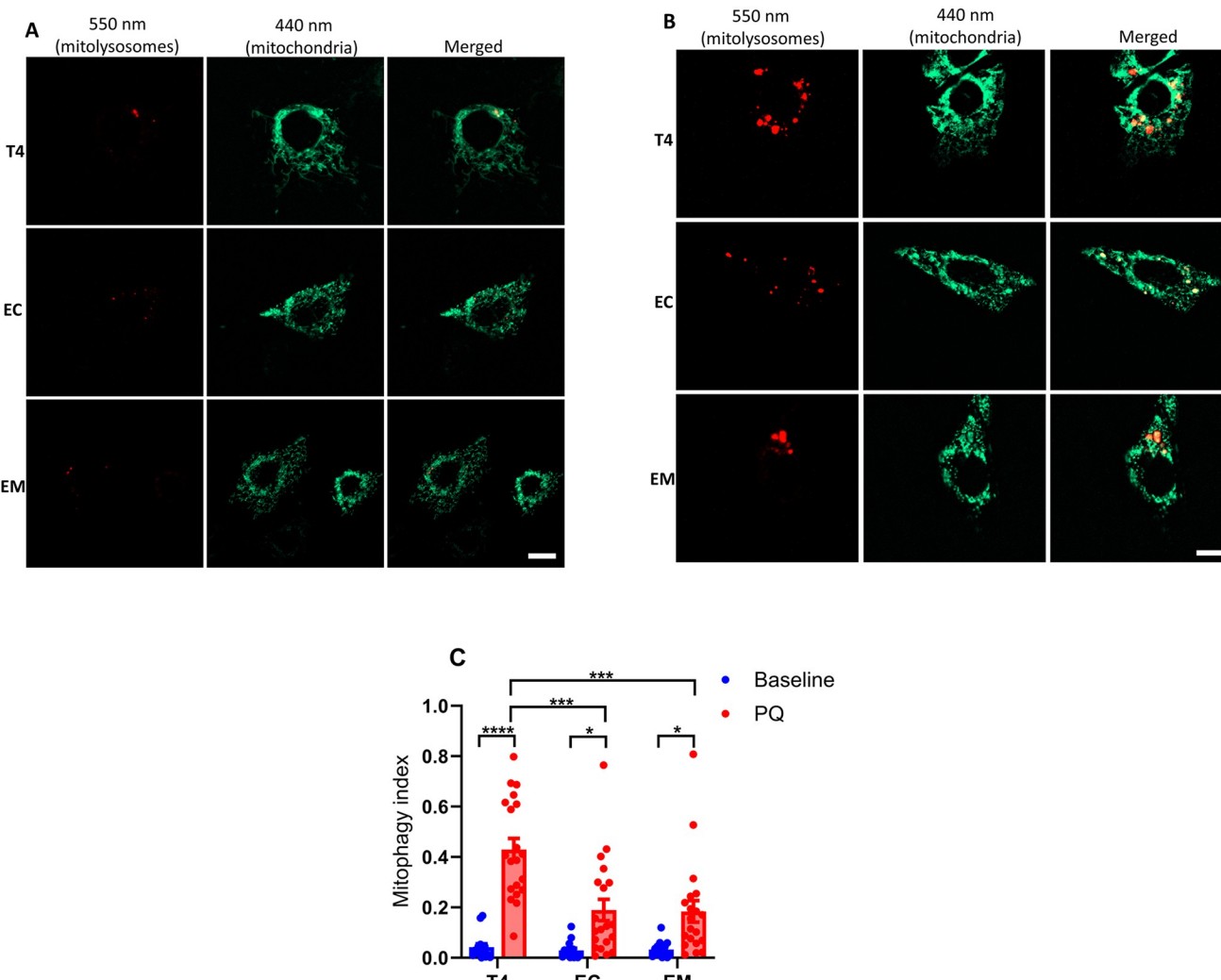

**Fig 2. Oxidative stress-induced mitophagy in HT22 cells.** Mitophagy was assessed in cells stably expressing mt-mKeima and respective tau constructs under unstressed conditions **(A)**, and after induction with 5 μM PQ for 6 h **(B)**. Fluorescence was measured using custom filter sets with excitation at 440 nm and 550 nm, and emission at 620 nm. Representative images were split into 550 nm (red: mitolysosome) and 440 nm (green: mitochondria) channels using ImageJ software. PQ exposure increased mitophagy, evidenced by increased red mitolysosome puncta. The mitophagy index, calculated as the ratio of the area and intensity of the mitolysosome (550 nm) to that of mitochondria (440 nm), was derived from three independent experiments. At least 5–7 cells per experiment were analyzed **(C)**. Data represent mean ± SEM and were statistically analyzed with two-way ANOVA followed by Tukey's post hoc test. $^*p < 0.05$; $^{**}p < 0.01$; $^{***}p < 0.001$; $^{****}p < 0.0001$. Scale bar: is 20 μm.

function as mitophagy receptors, facilitating the selective degradation of mitochondria [17]. PQ treatment significantly reduced FKBP8 receptor expression in cells expressing EC and EM tau isoforms but not in those expressing T4 (Fig 3D and 3E). Interestingly, FKBP8 expression remained relatively unchanged over time in T4-expressing cells, contrasting with the substantial degradation observed for other receptor proteins (Fig 3D and 3F). This persistence of FKBP8 in T4 cells suggests that pTau specifically impacts FKBP8 degradation, potentially inhibiting its normal function in mitophagy. In all tau-expressing groups, PQ treatment led to a significant decrease in the expression levels of both FUNDC1 and BNIP3 receptors (Fig 3D, 3G and 3H). Unlike FKBP8, these receptors were degraded consistently, indicating a broader impact of mitophagy induction on receptor-mediated pathways.

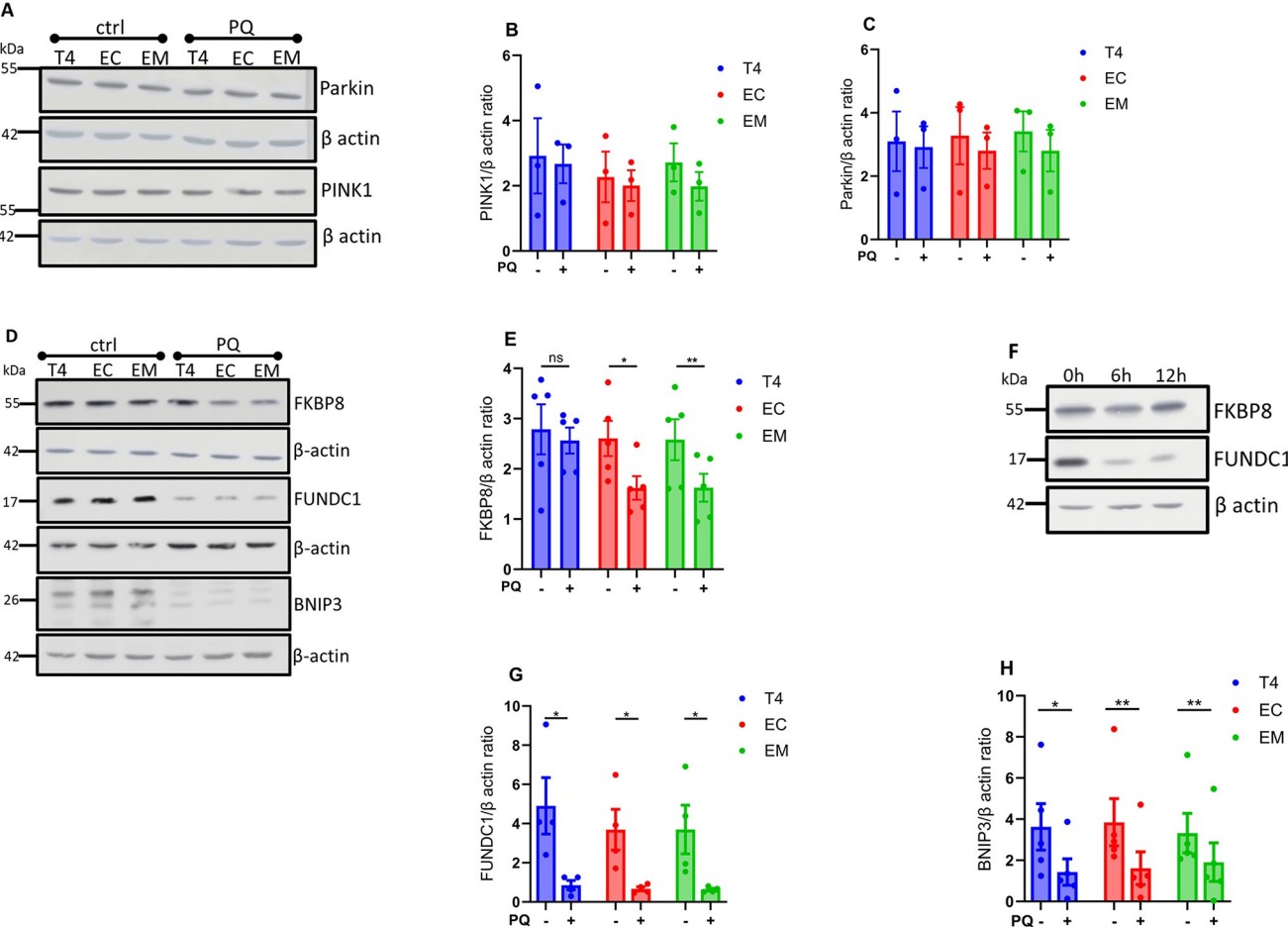

**Fig 3. Immunoblots analysis of mitophagy-related protein expression.** Cell lysates from unstressed (ctrl) cells and cells treated with PQ for 6 h to induce mitophagy were assessed using the indicated antibodies. Immunoblots of PINK1 and parkin protein expression (**A**). Densitometry quantification of PINK1 and parkin show no significant changes across all tau groups (**B, C**). Immunoblots for the indicated mitophagy receptors (**D**). Densitometry quantification of the respective mitophagy receptors show that FKBP8 expression was significantly reduced in cells expressing EC and EM compared to T4 (**E**), while BNIP3 and FUNDC1 receptor levels decreased significantly in all groups (**F, G**). Immunoblot shows changes in FKBP8 and FUNCD1 expression overtime (0 h, 6 h, 12 h) in cells expressing T4 (**H**). Proteins were normalized to their respective beta-actin and Ponceau-S. Data represent mean ± SEM from 3–5 independent experiments and were statistically analyzed with two-tailed Student's $t$-test. $*p < 0.05$; $**p < 0.01$.

These findings prompted us to probe whether FKBP8 interacts with tau. Hence, we conducted co-immunoprecipitation assays. Immunoblot analyses revealed that FKBP8 interacts with T4 as well as the phosphomimetics (Fig 4), though this assay does not determine if the binding is direct or requires other factors.

## Phosphorylated tau likely impacts FKBP8 dynamics at the mitochondrial-ER interface

Previous studies have shown that FKBP8 can evade degradation by translocating to the ER in a microtubule dependent manner during mitophagy [21, 42]. To investigate whether pTau impacts FKBP8 translocation, we used immunocytochemistry assay to assess its interaction with specific ER proteins during mitophagy. We analyzed the colocalization of FKBP8 with the ER specific protein BiP/GRP78 under unstressed conditions and following mitophagy induction. In the absence of stress, FKBP8 showed minimal colocalization with BiP/GRP78.

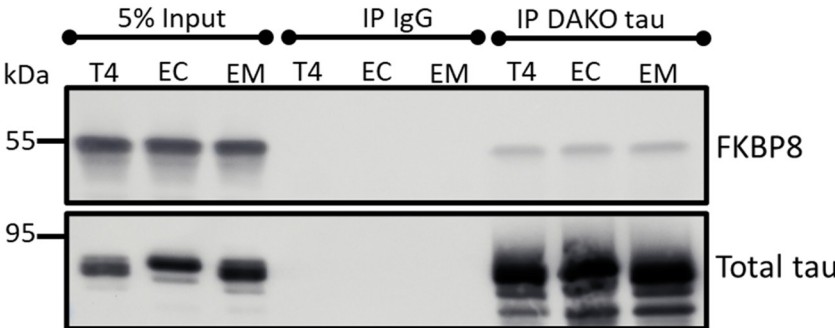

**Fig 4. Interaction of FKBP8 with tau.** Five hundred micrograms of lysate from cells expressing wildtype and phosphomimetic tau were immunoprecipitated with rabbit anti-tau (DAKO tau) or nonspecific rabbit IgG antibodies. Western blot analysis of the eluted proteins with mouse anti-FKBP8 shows co-immunoprecipitation of FKBP8 with tau. The blot was stripped and re-probed with mouse anti-tau antibodies (Tau5 and 5A6).

However, upon mitophagy induction, colocalization increased significantly in cells expressing wild-type tau (T4) and the tau isoform EC (Fig 5A–5C). Specifically, T4-expressing cells showed the most pronounced increase in FKBP8 colocalization with BiP/GRP78 (% increase = 66.5%, p = 0.0016), followed by EC-expressing cells (% increase = 55.7%, p = 0.0117). In contrast, cells expressing the EM did not exhibit a significant increase in colocalization (% increase = 31.7%, p = 0.0934). To further elucidate the impact of pTau on FKBP8 dynamics, we examined FKBP8 expression in crude mitochondrial fractions during mitophagy. Compared to the unstressed state, FKBP8 expression decreased significantly across all tau-expressing groups following mitophagy induction (Fig 5D and 5E). The most substantial decrease in FKBP8 levels was observed in T4-expressing cells (64.6%), followed by EC-expressing cells (58.1%) and EM-expressing cells (45.7%) (Fig 5F). Previous data showed that the reduction of FKBP8 in mitochondria crude fractions resulted in its increase in ER fraction during mitophagy [21]. Our findings, consistent with other reports, highlight FKBP8 dynamics at the mitochondrial-ER interface. Specifically, our results indicate that FKBP8 translocation to the ER appears to protect it from degradation -a protective mechanism that may be suppressed in tauopathies.

Furthermore, we tested whether microtubule destabilization with nocodazole could decrease FKBP8 expression in T4 cells, potentially highlighting previously described microtubule-dependent translocation to the ER [21] and elucidating the reduced FKBP8 expression in cells expressing phosphomimetic tau. We found that nocodazole treatment for 6 h impaired cell projections and cell morphology in all tau groups (S4A Fig) but did not alter FKBP8 expression trends relative to PQ exposure (S4B Fig).

### Assessing mitophagy in FKBP8 knockdown cells

To estimate the contribution of FKBP8 receptor to oxidative stress-induced mitophagy, we depleted endogenous FKBP8 in our stable cell lines expressing mt-mKeima using lentivirus infection. The efficacy of knockdown determined by western blotting shows that the protein levels of FKBP8 were robustly decreased in cell lysates (Fig 6A). Knockdown of FKBP8 also resulted in a decrease in total tau levels in the T4 cells, but not in EC- or EM-expressing cells (S6 Fig).

The knockdown cells were then treated with PQ for 6 h before microscopic analysis. Our findings reveal that FKBP8 knockdown led to an insignificant decrease in mitophagy across all

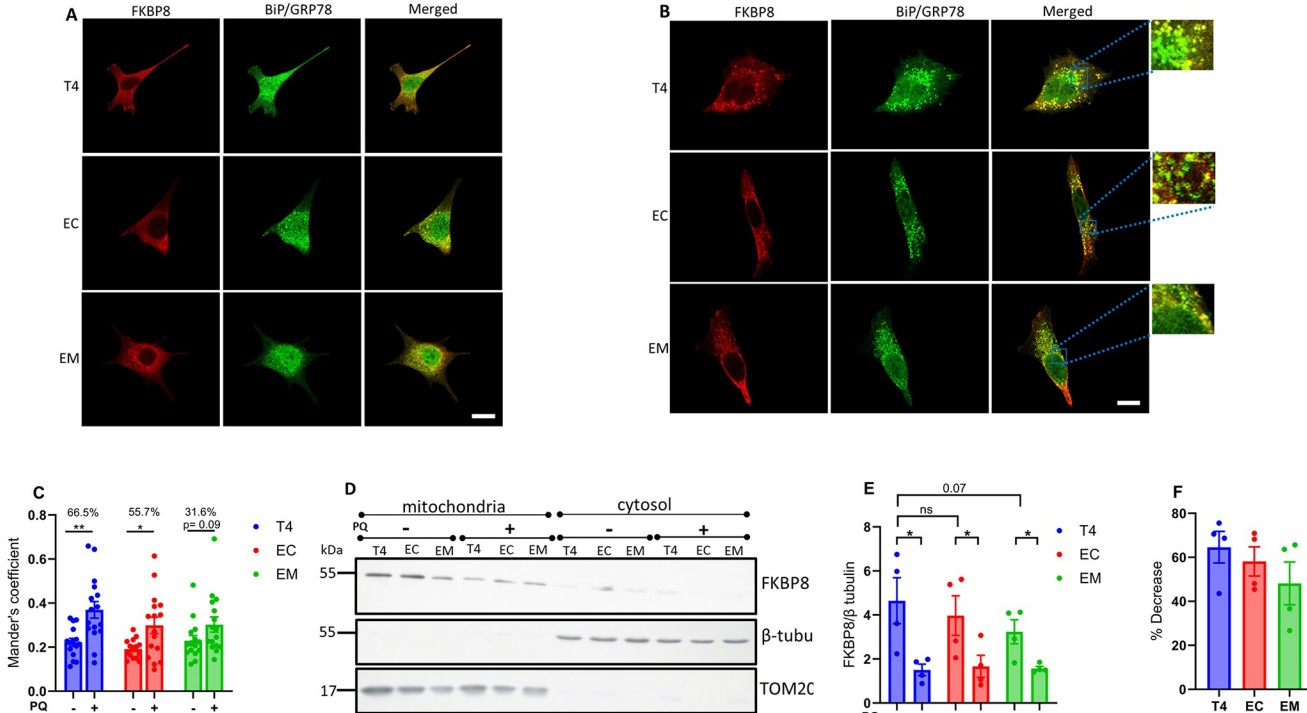

**Fig 5. Interaction of FKBP8 with ER specific protein and FKBP8 expression in mitochondria fraction during mitophagy.** Dual staining for FKBP8 (red) and BiP/GrP78 (green) was performed to analyze their colocalization in unstressed cells (**A**) and cells induced for mitophagy with PQ for 6 h (**B**). Representative immunofluorescence merged images from confocal microscopy reveal pronounced colocalization of these proteins during mitophagy. Magnified regions of the merged images are shown in the blue box. Mander's coefficient of colocalization (**C**) was calculated from three independent experiments using Imaris 10.0 program (Oxford Instruments). Crude mitochondrial and cytosolic fractions were isolated by differential centrifugation from cells expressing wildtype and phosphomimetic tau. Equal protein amounts were subjected to Western blot analysis with an anti-FKBP8 antibody. β-tubulin (cytoplasmic marker) and TOM20 (mitochondrial outer membrane marker) were used to verify fraction purity. Representative immunoblot (**D**), densitometry analysis (**E**), and percentage decrease estimation (**F**) show reduced mitochondrial FKBP8 expression across all tau groups. Quantification is expressed as the FKBP8/β-tubulin ratio and represents mean values ± SEM from four independent experiments. Statistical analysis was performed using two-tailed Student's *t*-test. $^*p < 0.05$; $^{**}p < 0.01$. Scale bar: 25 μm.

tau groups compared to scramble (Fig 6B and 6C). The % decreases were 21.0, 15.5 and 13.2 for T4 (p = 0.401), EC (p = 0.752) and EM (p = 0.779), respectively. This finding suggests that FKBP8 directly contributes minimally to oxidative stress-induced mitophagy, and that other pathways likely compensate for mitophagy in FKBP8 knockdown cells.

## Discussion

The coexistence of pTau and accumulated damaged mitochondria in AD is thought to result in part from defective mitophagy [3, 43]. However, the molecular mechanisms through which pTau impacts the mitophagy cascade remain elusive. Identifying the specific molecular interactor(s) and understanding how pTau disrupts mitophagy could provide valuable insights into AD pathogenesis and potential therapeutic targets. Our study contributes to understanding the roles of pTau in AD by demonstrating that pTau, particularly at Thr231/Ser235 and Ser396/404 residues, suppresses oxidative stress-induced mitophagy. This finding not only corroborates our previous report in *C. elegans* [27] but extends it to mammalian cells. We identified FKBP8 receptor as a plausible target that reports pTau toxicity and propose a mechanism whereby pTau alters FKBP8 dynamics and attenuates its contributions to overall mitophagy process.

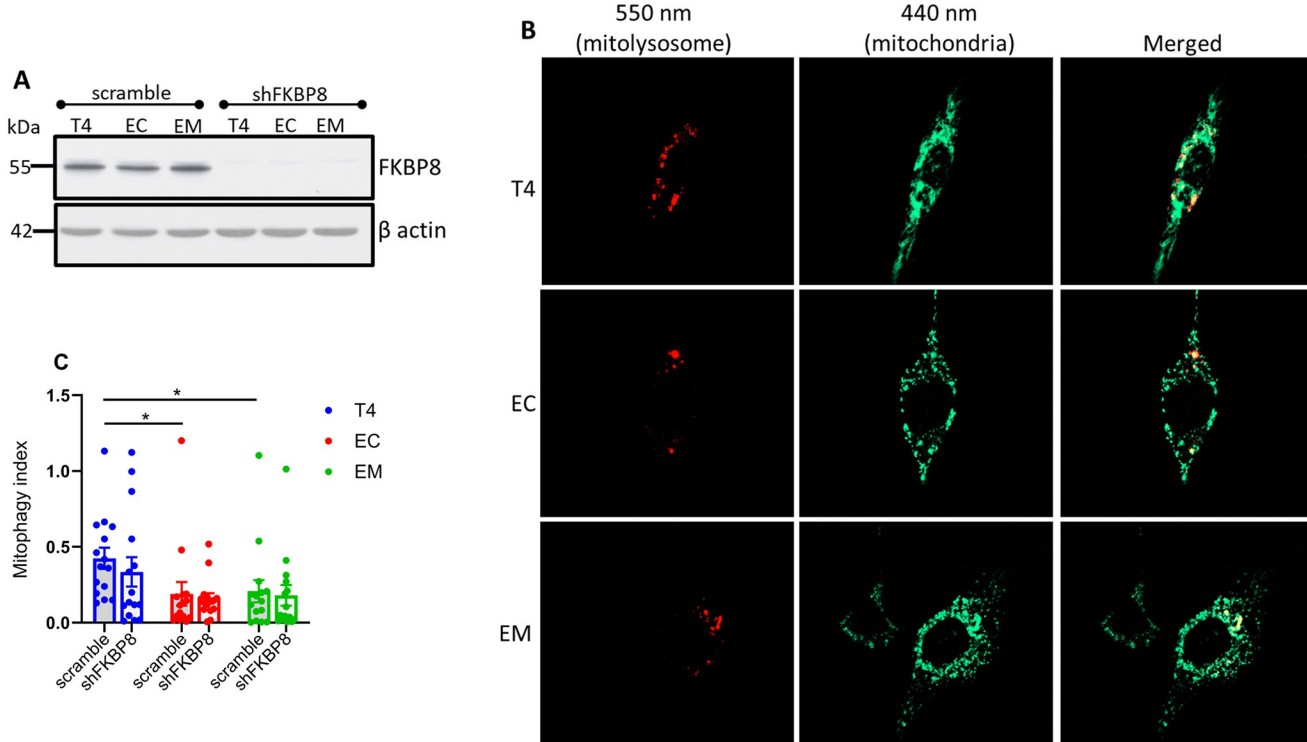

**Fig 6. Mitophagy in FKBP8 knockdown cells.** FKBP8 was knocked down in stable cell lines, confirmed by Western blotting **(A).** Mitophagy was assessed after induction with 5 μM PQ for 6 h. Fluorescence was measured using custom filter sets with excitation at 440 nm and 550 nm, and emission at 620 nm. Representative images were split into 550 nm (red: mitolysosome) and 440 nm (green: mitochondria) channels using ImageJ software **(B).** Mitophagy index calculated from three independent experiments and compared to stable cells treated with PQ for 6 h show that FKBP8 knockdown did not significantly suppress oxidative stress-induced mitophagy in all tau groups **(C).** Data represent mean ± SEM and were statistically analyzed using Student's $t$-test. *$p < 0.05$. Scale bar: 20 μm.

The pathogenesis of AD is influenced by genetic predispositions, impaired proteostasis, and exposure to environmental stressors such as PQ [44, 45]. PQ induces mitophagy by interacting with redox sites in the mitochondrial electron transport system, leading to ROS generation and oxidative stress [46, 47]. Exposure of cells to a low concentration of PQ (5 μM) for 6 h stimulates ROS generation and induces mitophagy, whereas prolonged exposure caused severe cellular damage and abnormal morphology (S5 Fig). The attenuated oxidative stress-induced mitophagy in cells expressing EC and EM provides evidence that mitophagy dysregulation is likely one of the drivers of pTau toxicity in AD pathology [40]. Notably, impaired mitophagy possibly elucidates the excessive ROS accumulation we previously reported in these cells [36], as well as the oxidative damage induced by EC in AD patients' brain [26]. Our findings that PINK1/parkin expression levels remained largely unchanged across all tau groups during mitophagy, while FUNDC1 and BNIP3 levels were markedly reduced, suggest that receptor-mediated mitophagy is the first line mechanism of mitochondrial quality control against oxidative stress in our model. The importance of the receptor-mediated mitophagy during oxidative stress has been described [48] and underscored by the finding that PINK1 knockdown stimulates parkin-independent oxidative stress-induced mitophagy in SH-SY5Y cells [49]. In accordance, FKBP8 receptor-mediated mitophagy has been proposed as an important sensor of mitochondrial stress-related stimuli for mitophagy [17].

The FKBPs act as receptors of FK506, featuring prolyl cis/trans isomerase (PPIase) and tetratricopeptide (TCP) domains [50, 51]. The PPIase domain mediates immunosuppressive

activities and protein folding, while the TCP domain interacts with cellular proteins, crucial for protein trafficking [52, 53]. They are highly expressed in the brain but show reduced expression in AD patients [24, 54, 55]. FK506 has been shown to decrease tau aggregation and increase the lifespan of P301S mice, suggesting an important role of FKBPs in AD pathology [56]. Unlike other FKBPs, FKBP8 lacks constitutive PPIase activity, suggesting involvement in other cellular functions such as neuroprotection and neurite outgrowth [24, 50]. FKBP8 is unique in the FKBP family as it contains a transmembrane domain, anchoring it to the mitochondria where it functions as a mitophagy receptor [16, 21, 57]. Previous studies have elucidated the roles of FKBP8 in stress-induced mitophagy [16, 21, 42]. To our knowledge, we are the first to demonstrate the interaction between pTau and FKBP8 and its impact on FKBP8 receptor-mediated mitophagy. By demonstrating that phosphomimetic tau suppresses mitophagy, which corresponds with decreased expression of FKBP8, we provide insight that pTau likely alters FKBP8's contribution to mitophagy.

Our observation that mitophagy increases the colocalization of FKBP8 and ER-resident proteins, and reduces FKBP8 expression in isolated crude mitochondria, aligns with previous findings in SH-SY5Y cells and mouse embryonic fibroblasts [21]. While further molecular studies are needed to dissect the mitochondrial-ER-FKBP8 dynamics, our results provide additional evidence of FKBP8 translocation to the ER during mitophagy consistent with prior reports [16, 21, 42] and suggest that pTau likely impacts the translocation process. Indeed, our previous research and others [2, 30, 33] have established that pTau disrupts microtubule stabilization and impairs cellular transport of proteins and organelles. Therefore, it is plausible that EC and EM mutations disrupt microtubules, thereby hampering FKBP8 transport. These suggest a link between microtubule dynamics, tau phosphorylation, FKBP8 transport, and mitophagy regulation. Further investigation into these interconnected pathways promises to deepen our understanding about cellular protein transport during mitophagy and pave the way for the development of targeted interventions to combat mitochondrial quality control-associated diseases. Contrary to immunocytochemistry analysis that revealed that nocodazole treatment impair microtubule translocation to the ER [21], our Western blot assay showed no effect of microtubule destabilization with nocodazole on FKBP8 expression (S4B Fig). This discrepancy is likely due to the Western blot's limited ability to determine the cellular location of the protein.

Discrepancies exist regarding the effect of FKBP8 knockdown on stress-induced mitophagy. Yoo et al. showed that hypoxia- and iron deficiency-induced mitophagy were significantly suppressed by FKBP8 knockdown in HeLa cells and human fibroblasts [42]. However, knockdown of FKBP8 had no effect on CCCP-induced mitophagy in H9c2 myocytes or HEK293 cells but resulted in elevated levels of misfolded proteins and ER-stress markers [58]. Variation in stressor type, concentration, duration of exposure, and cell type can account for the inconsistencies. These also highlight the intricate nature and complexity of mitophagy regulation. Nevertheless, both studies concluded that FKBP8 plays crucial roles in stress-induced mitophagy [42, 58]. Here, the decreased mitophagy we found in FKBP8 knockdown cells was not statistically significant. There are possible explanations. First, knockdown of FKBP8 may trigger compensatory mechanisms that upregulate mitophagy through other pathways.

Studies in *Drosophila* and HeLa cells [59] demonstrated that mitochondrial E3 ubiquitin protein ligase 1 (MUL1) can mediate mitophagy in the absence of PINK1/Parkin by regulating mitofusin levels. Similarly, in *C. elegans*, knockdown of DCT-1, a mitophagy receptor, or PINK1 led to the upregulation of SKN-1, a transcription factor involved in stress responses [60]. Evidence of compensatory mitophagy response has been demonstrated in PINK1-PRKN knockout mouse [12], further supporting the existence of compensatory pathways. These findings suggest that FKBP8 knockdown may also induce alternative mitophagy mechanisms,

potentially through upregulation of other mitophagy receptors or stress-response proteins. Understanding these compensatory mechanisms will be crucial for fully elucidating the role of FKBP8 in mitophagy regulation.

Alternatively, due to molecular remodeling and gain of function imposed by phosphorylation of tau [61], the cells may have developed the intrinsic ability to eliminate damaged mitochondria by bulk autophagy when FKBP8 receptor is depleted. Third, the involvement of accessory proteins that steer FKBP8 functions suggest that the receptor may not be the primary target. Proteins such as Bcl2 and Hsp90 are required for FKBP8's anti-apoptotic and chaperone functions, respectively. Indeed, tau has been shown to interact with Hsp90, resulting in conformational changes [62, 63]. Although the specific protein coupled with FKBP8 during mitophagy remains elusive, reports indicate that FKBP8 interacts with OPA1 [42] and prohibitin [48] to mediate mitophagy. Hence, it is plausible that one or more of these proteins are the main targets of pTau toxicity, reflected by decreased FKBP8 expression. This reduced expression of FKBP8 may serve as a surrogate readout indicating the system impacted by pTau, possibly along the mitochondrial-ER axis. Indeed, as a mobile mitochondria-associated ER membrane (MAM) protein, FKBP8 regulates $Ca^{2+}$ transport from the ER to the mitochondria and plays crucial roles in MAM formation under stress conditions [64, 65]. Therefore, the decreased expression in EC and EM possibly indicates impaired $Ca^{2+}$ homeostasis and increased susceptibility to ER-Ca stress, as we have previously shown [36, 66].

Given the significance of impaired mitophagy in AD pathogenesis, identifying and regulating specific mediators of mitophagy pathways could be crucial in AD therapy. The neuroprotective effect of FK506 and the ability of FKBPs to protect against tau toxicity have been demonstrated in *in vitro*, *in vivo*, and animal models of neurodegenerative diseases [45, 52, 67]. Therefore, FKBP8 might represent a unique potential target for mitochondrial quality control and tau toxicity ablation in AD pathogenesis.

Alternatively, accumulating evidence suggests that tau deletion may have neuroprotective effects, particularly on mitochondrial function and aging [68, 69]. Our previous work [70] and Jara et al. [69, 71] showed that genetic ablation of tau protects cortical neurons from $A\beta_{42}$-induced mitochondrial membrane potential loss and improved mitochondrial health in aging mice, respectively. This effect extends to preventing neurotoxic insults linked to overactivation of extrasynaptic NMDA receptors [72]. These findings suggest that tau depletion may prevent aging-related pTau accumulation and mitophagy dysregulation. Future research on how tau ablation impacts basal and stress-induced mitophagy could provide valuable insights into the therapeutic potential of tau reduction in age-related tauopathies involving mitochondrial dysfunction.

## Supporting information

**S1 Fig. Receptor-mediated mitophagy.** Mitophagy receptors on the outer mitochondrial membrane, including FKBP8, BNIP3, FUNDC1, and NIX, detect damaged mitochondria and recruit the phagophore. These receptors interact with LC3 on the phagophore through their LC3-interacting region (LIR), facilitating the engulfment of the damaged organelle and the formation of a mitophagosome. The mitophagosome then fuses with a lysosome for degradation. Created with BioRender.com.
(TIF)

**S2 Fig. Tau expression in mouse neurons and stable HT22 cell lines.** Cell lysates (20 μg protein) prepared from primary mouse neurons and stable HT22 cells expressing wildtype and phosphomimetic tau were analyzed by Western blot using anti-tau antibody (DAKO tau: 1:20000). Immunoblot shows that tau expression levels in primary mouse neurons were

markedly higher compared to all tau groups. The control lane (ctrl) confirms the absence of endogenous tau in HT22 cells.
(TIF)

**S3 Fig. Mitochondrial labeling with MitoTracker CMX-Ros.** Representative images of cells pre-treated with paraquat (PQ) and subsequently incubated with MitoTracker CMX-Ros (20 nM) for 15 minutes. MitoTracker CMX-Ros selectively accumulates in active mitochondria, labeling them with red fluorescence. Colocalization showing overlap between mt-mKeima signal and MitoTracker CMX-Ros fluorescence confirms the targeting and localization of mt-mKeima to the mitochondria. Scale bars represent 20 μm.
(TIF)

**S4 Fig. Nocodazole treatment alters cell morphology but not FKBP8 expression.** Cells were incubated with 5 μM nocodazole for 6 h to explore whether microtubule destabilization influences FKBP8 expression. Images taken under brightfield acquisition show that nocodazole treatment impaired cell projections and cell morphology **(A)**. However, Western blot analysis shows that FKBP8 expression trends remain unchanged compared to PQ exposure **(B)**.
(TIF)

**S5 Fig. Prolonged exposure to paraquat caused cellular damage.** Representative images showing excess and clumped mitolysosome, highlighting marked cellular damage and altered cell morphology following 12 h exposure to PQ (5 μM). Scale bar: is 15 μm.
(TIF)

**S6 Fig. Tau expression reduced in FKBP8 knockdown cells.** Cell lysates prepared from FKBP8 knockdown and scramble cells were analyzed by Western blot for total tau (Dako tau, A0024). Immunoblot shows that tau expression levels decreased prominently in FKBP8 knockdown cells expressing wildtype tau (T4) compared to scramble.
(TIF)

**S1 Raw images.**
(PDF)

## Acknowledgments

We extend our gratitude to Dr. Hemachandra Reddy from Texas Tech University for generously providing the HT22 cells, and to Dr. Christoph Pröschel from the University of Rochester for supplying the lentiviral vectors. We also thank Dr. Peter Davies for the phospho-Ser396/404 antibody. The diagram in this study was prepared by BioRender (https://www.biorender.com/).

## Author Contributions

**Conceptualization:** Michael O. Isei, Keith Nehrke, Gail V. W. Johnson.

**Data curation:** Meredith Crockett, Joel Rodwell-Bullock, Keith Nehrke.

**Formal analysis:** Michael O. Isei, Meredith Crockett, Joel Rodwell-Bullock, Keith Nehrke, Gail V. W. Johnson.

**Funding acquisition:** Gail V. W. Johnson.

**Investigation:** Michael O. Isei, Joel Rodwell-Bullock, Trae Carroll, Peter A. Girardi, Keith Nehrke, Gail V. W. Johnson.

**Methodology:** Michael O. Isei, Meredith Crockett, Emily Chen, Joel Rodwell-Bullock, Trae Carroll, Peter A. Girardi, Keith Nehrke, Gail V. W. Johnson.

**Supervision:** Michael O. Isei.

**Writing – original draft:** Michael O. Isei, Keith Nehrke, Gail V. W. Johnson.

**Writing – review & editing:** Michael O. Isei, Meredith Crockett, Emily Chen, Joel Rodwell-Bullock, Trae Carroll, Peter A. Girardi, Keith Nehrke, Gail V. W. Johnson.

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
