## [Decision Letter · Decision Letter 0]

17 Sep 2024

PONE-D-24-27298Tau phosphorylation suppresses oxidative stress-induced mitophagy via FKBP8 receptor modulationPLOS ONE

Dear Dr. Johnson,

Thank you for submitting your manuscript to PLOS ONE. After careful consideration, we feel that it has merit but does not fully meet PLOS ONE’s publication criteria as it currently stands. Therefore, we invite you to submit a revised version of the manuscript that addresses the points raised during the review process.

Reviewer 1: In this work, Isei et al. have shown that phosphorylated tau at specific residues decreases oxidative stress-induced mitophagy. In previous studies, the group described that mutated tau (Thr231E) inhibits mitophagy, now they found that also tau phosphorylation at other residues suppresses oxidative-induced mitophagy, through a novel mechanism, the interaction of phospho tau to FKBP8 receptor. The work is sound but some specific points should be discussed.

- There are several factors, like phosphorylated tau, parkin or FKBP8 receptor that could affect mitochondrial function. About tau, it should be indicated not only the gain of (dys)function by phosphor tau, but also that the absence of tau could facilitate mitochondrial action (see for example Front. Neurosc. 2021, 14: 586710). Also, it was shown that modified tau impairs mitophagy via parkin inhibition (see for example reference 9). Now, in this work, a new role for tau in mitophagy based on its binding to FKBP8 receptor has been clearly indicated. In summary, it should be focused the role of tau by different ways on mitophagy regulation. The roles of tau or parkin have been extensively shown. However, it is reported that FKBP8 knock down does not affect to mitophagy. It is suggested that a compensatory mechanism could upregulate mitophagy through other pathways. Perhaps, some of those pathways should be commented. Also, k.o. of FKBP8 does not have effect in CCCP-induced mitophagy in HEK 293 cells and it was suggested that it may be due to the stressor type used. In this way, for some experiments, it could be advisable to use two different stressor types.

- The data of Figure S3B may be complemented with immunochemistry analysis if a suitable antibody is available.

Minor point:

Eight Figures are many figures, perhaps Figure 1 could be a supplementary Figure and Figures 6 and 7 may be joined in a single one.

We look forward to receiving your revised manuscript.

Kind regards,

Vladimir Trajkovic

Academic Editor

PLOS ONE

Journal Requirements:

 1. When submitting your revision, we need you to address these additional requirements. Please ensure that your manuscript meets PLOS ONE's style requirements, including those for file naming. The PLOS ONE style templates can be found at https://journals.plos.org/plosone/s/file?id=wjVg/PLOSOne_formatting_sample_main_body.pdf and https://journals.plos.org/plosone/s/file?id=ba62/PLOSOne_formatting_sample_title_authors_affiliations.pdf. 2. We note that the grant information you provided in the ‘Funding Information’ and ‘Financial Disclosure’ sections do not match.  When you resubmit, please ensure that you provide the correct grant numbers for the awards you received for your study in the ‘Funding Information’ section. 3. Thank you for stating the following financial disclosure:  [NIH R01 AG067617].  Please state what role the funders took in the study.  If the funders had no role, please state: ""The funders had no role in study design, data collection and analysis, decision to publish, or preparation of the manuscript."" If this statement is not correct you must amend it as needed. Please include this amended Role of Funder statement in your cover letter; we will change the online submission form on your behalf. 4. Thank you for stating the following in the Acknowledgments Section of your manuscript: [We extend our gratitude to Dr. Hemachandra Reddy from Texas Tech University for generously providing the HT22 cells, and to Dr. Christoph Pröschel from the University of Rochester for supplying the lentiviral vectors. We also thank Dr. Peter Davies for the phospho-Ser396/404 antibody. This work was supported by the National Institutes of Health (NIH) R01 AG067617.The diagram in this study was prepared by BioRender (https://www.biorender.com/).]We note that you have provided funding information that is not currently declared in your Funding Statement. However, funding information should not appear in the Acknowledgments section or other areas of your manuscript. We will only publish funding information present in the Funding Statement section of the online submission form. Please remove any funding-related text from the manuscript and let us know how you would like to update your Funding Statement. Currently, your Funding Statement reads as follows:  [NIH R01 AG067617]. Please include your amended statements within your cover letter; we will change the online submission form on your behalf. 5. PLOS ONE now requires that authors provide the original uncropped and unadjusted images underlying all blot or gel results reported in a submission’s figures or Supporting Information files. This policy and the journal’s other requirements for blot/gel reporting and figure preparation are described in detail at https://journals.plos.org/plosone/s/figures#loc-blot-and-gel-reporting-requirements and https://journals.plos.org/plosone/s/figures#loc-preparing-figures-from-image-files. When you submit your revised manuscript, please ensure that your figures adhere fully to these guidelines and provide the original underlying images for all blot or gel data reported in your submission. See the following link for instructions on providing the original image data: https://journals.plos.org/plosone/s/figures#loc-original-images-for-blots-and-gels.   In your cover letter, please note whether your blot/gel image data are in Supporting Information or posted at a public data repository, provide the repository URL if relevant, and provide specific details as to which raw blot/gel images, if any, are not available. Email us at plosone@plos.org if you have any questions. 6. Please include captions for your Supporting Information files at the end of your manuscript, and update any in-text citations to match accordingly. Please see our Supporting Information guidelines for more information: http://journals.plos.org/plosone/s/supporting-information. 

Reviewers' comments:

Reviewer's Responses to Questions

**Comments to the Author**

1. Is the manuscript technically sound, and do the data support the conclusions?

Reviewer #1: Yes

2. Has the statistical analysis been performed appropriately and rigorously? 

Reviewer #1: Yes

3. Have the authors made all data underlying the findings in their manuscript fully available?

Reviewer #1: Yes

4. Is the manuscript presented in an intelligible fashion and written in standard English?

Reviewer #1: Yes

5. Review Comments to the Author

Reviewer #1: In this work, Isei et al. have shown that phosphorylated tau at specific residues decreases oxidative stress-induced mitophagy. In previous studies, the group described that mutated tau (Thr231E) inhibits mitophagy, now they found that also tau phosphorylation at other residues suppresses oxidative-induced mitophagy, through a novel mechanism, the interaction of phospho tau to FKBP8 receptor. The work is sound but some specific points should be discussed.

- There are several factors, like phosphorylated tau, parkin or FKBP8 receptor that could affect mitochondrial function. About tau, it should be indicated not only the gain of (dys)function by phosphor tau, but also that the absence of tau could facilitate mitochondrial action (see for example Front. Neurosc. 2021, 14: 586710). Also, it was shown that modified tau impairs mitophagy via parkin inhibition (see for example reference 9). Now, in this work, a new role for tau in mitophagy based on its binding to FKBP8 receptor has been clearly indicated. In summary, it should be focused the role of tau by different ways on mitophagy regulation. The roles of tau or parkin have been extensively shown. However, it is reported that FKBP8 knock down does not affect to mitophagy. It is suggested that a compensatory mechanism could upregulate mitophagy through other pathways. Perhaps, some of those pathways should be commented. Also, k.o. of FKBP8 does not have effect in CCCP-induced mitophagy in HEK 293 cells and it was suggested that it may be due to the stressor type used. In this way, for some experiments, it could be advisable to use two different stressor types.

- The data of Figure S3B may be complemented with immunochemistry analysis if a suitable antibody is available.

Minor point:

Eight Figures are many figures, perhaps Figure 1 could be a supplementary Figure and Figures 6 and 7 may be joined in a single one.

6. PLOS authors have the option to publish the peer review history of their article (what does this mean?). If published, this will include your full peer review and any attached files.

Reviewer #1: **Yes: **Jesus Avila

---

## [Author Response · Author response to Decision Letter 0]

9 Oct 2024

PONE-D-24-27298

“Tau phosphorylation suppresses oxidative stress-induced mitophagy via FKBP8 receptor modulation”

To the Editor: 

We thank the editor for overseeing the review of our manuscript and the opportunity to revise it. The reviewer’s comments are in italics and our responses in regular font.

To reviewer 1:

We thank the reviewer for their careful review and appreciation of our work. Their comments and suggestions are very thoughtful and useful, and we have addressed them on a point-by-point basis as outlined below. 

1. “About tau, it should be indicated not only the gain of (dys)function by phosphor tau, but also that the absence of tau could facilitate mitochondrial action (see for example Front. Neurosc. 2021, 14: 586710)”.

Response: We appreciate the reviewer for bringing this important point to our attention and believe this addition will strengthen our manuscript. We agree that it is important to expand our manuscript to address that coupled with the gain of (dys)function imposed by phosphorylated tau, the absence of tau could facilitate mitochondrial action.

We have now discussed our previous work and other findings indicating that tau depletion/knockout may have neuroprotective effects, particularly on mitochondrial function and aging (Pallas-Bazarra et al., 2019, PMID:31235881; Pallo & Johnson, 2015, PMID:25888814; Jara et al., 2020, PMID:33679286; Jara et al., 2018, PMID:30077079). We also highlighted that tau depletion may prevent aging-related accumulation of phosphorylated tau and mitophagy dysregulation (page 24, line 520-529 in the manuscript). We also added a paper that explored the interplay between tau phosphorylation and Alzheimer’s disease (Avila, 2006, PMID:16529745)

2. “The roles of tau or parkin have been extensively shown. However, it is reported that FKBP8 knock down does not affect to mitophagy. It is suggested that a compensatory mechanism could upregulate mitophagy through other pathways. Perhaps, some of those pathways should be commented”

Response: We thank the reviewer for the suggestion. We have now included a discussion on relevant studies highlighting such compensatory mechanisms. For example, studies in Drosophila and HeLa cells (Yun et al., 2014, PMID:24898855) demonstrated that MUL1-mediated mitophagy can compensate for the loss of PINK1/Parkin by regulating mitofusin levels. Similarly, in C. elegans, knockdown of DCT-1 (a mitophagy receptor; BNIP3 in vertebrates) or PINK1 results in upregulation of SKN-1 (Nrf2 in vertebrates)-mediated mitophagy (Palikaras et al., 2015, PMID:25896323). These examples provide insights into how alternative mitophagy pathways may be activated when FKBP8 is knocked down, ensuring mitochondrial quality control. (Page 23, line 488-497 in the manuscript)

3. Also, k.o. of FKBP8 does not have effect in CCCP-induced mitophagy in HEK 293 cells and it was suggested that it may be due to the stressor type used. In this way, for some experiments, it could be advisable to use two different stressor types.

Response: We thank the reviewer for the insightful comment. In this study, besides Paraquat, we also investigated the effect of Urolithin A on mitophagy in our model, and unexpectedly, it did not significantly stimulate mitophagy. The insignificant mitophagy induction observed with Urolithin A in our cell line underscores the complexity and context-dependency of mitophagy regulation. This result, while initially surprising, aligns with the growing body of evidence suggesting that mitophagy induction is highly dependent on the specific stressor, cell type, and organism involved. Furthermore, the efficacy of Urolithin A may depend on specific genetic factors or signaling pathways present in certain cell types but not others. Our cell line may lack certain receptors or signaling components necessary for Urolithin A-mediated mitophagy activation.

Our group is currently exploring the effects of three additional stressors, each with distinct mechanisms of action, on mitophagy in C. elegans. These studies aim to understand how different stressors induce mitophagy and the potential compensatory pathways involved. We are also extending this investigation to our newly acquired immortalized human neural progenitor cell line (ReN cells), which can differentiate into neurons and glial cells, allowing us to test these stressors in a more relevant human model. This approach will provide deeper insights into the variability of mitophagy responses across different cell types and stressor conditions.

4. The data of Figure S3B may be complemented with immunochemistry analysis if a suitable antibody is available.

Response: We appreciate the reviewer's suggestion. Previously, we have tried to assess microtubule dynamics upon treatment with nocodazole but encountered some challenges. We attempted to use a suitable antibody for immunochemistry analysis of microtubules upon exposure to nocodazole. We found that our antibodies lack the specificity and sensitivity required for reliable immunochemistry analysis in our experimental setup. To overcome this limitation, we developed an alternative approach. We generated a BFP:FKBP8:Tau plasmid, which will allows us to monitor FKBP8 dynamics along the microtubules through dynamic fluorescent imaging in real-time using confocal microscopy. However, we encountered a significant challenge with this approach. Due to the spectral properties of the fluorophores involved in our experimental setup (BFP:CFP), we observed considerable spectral bleed-through leading to false positive signals and reduced data reliability.

5. Minor point: Eight Figures are many figures, perhaps Figure 1 could be a supplementary Figure and Figures 6 and 7 may be joined in a single one

Response: We thank the reviewer for this suggestion. 

Figure 1 has been moved to the supplementary materials as Supplementary Figure S1. As suggested, we have merged Figures 6 and 7, now labeled as Figure 5 in the revised manuscript. With these changes, we have reduced the total number of main figures from eight to six. We have updated all figure references throughout the text to reflect these changes. The figure legends have been revised to accommodate the merged figure, ensuring clear explanation of all data presented.

Journal Requirements:

Response: We have revised the manuscript to align with the style requirements and templates. Specifically, we adjusted the font sizes of the major sections and subsections, added asterisks to indicate the corresponding authors, and removed physical addresses, leaving only their email addresses. The changes have been highlighted for your reference.

Response: Thank you for the observation. The funding information is NIH R01 AG067617. 

Response: Thank you. 

Response: ""The funders had no role in study design, data collection and analysis, decision to publish, or preparation of the manuscript."" 

[We extend our gratitude to Dr. Hemachandra Reddy from Texas Tech University for generously providing the HT22 cells, and to Dr. Christoph Pröschel from the University of Rochester for supplying the lentiviral vectors. We also thank Dr. Peter Davies for the phospho-Ser396/404 antibody. This work was supported by the National Institutes of Health (NIH) R01 AG067617.

The diagram in this study was prepared by BioRender (https://www.biorender.com/).]

 [NIH R01 AG067617].

Response: Thank you for appreciating our acknowledgement. We have now removed the funding information from the acknowledgement section and other area in the manuscript. The funding statement is, NIH R01 AG067617.

Response: All uncropped and unadjusted images underlying all blot or gel results reported in a submission’s figures or Supporting Information files are included in Supporting Information as: S1_raw_images.

Response: We have now included title and captions of the supporting information files at the end of the manuscript, and update in-text citations accordingly.

Response: We have reviewed the reference list and highlighted the six newly added references.

Thank you very much for considering our submission. We believe that these revisions have substantially improved the manuscript, and we hope that it is now acceptable for publication in PLOS ONE journal.

Sincerely,

Gail VW Johnson

---

## [Decision Letter · Decision Letter 1]

21 Oct 2024

Tau phosphorylation suppresses oxidative stress-induced mitophagy via FKBP8 receptor modulation

PONE-D-24-27298R1

Dear Dr. Johnson,

We’re pleased to inform you that your manuscript has been judged scientifically suitable for publication and will be formally accepted for publication once it meets all outstanding technical requirements.

Kind regards,

Vladimir Trajkovic

Academic Editor

PLOS ONE

Additional Editor Comments (optional):

Reviewers' comments:

Reviewer's Responses to Questions

**Comments to the Author**

1. If the authors have adequately addressed your comments raised in a previous round of review and you feel that this manuscript is now acceptable for publication, you may indicate that here to bypass the “Comments to the Author” section, enter your conflict of interest statement in the “Confidential to Editor” section, and submit your "Accept" recommendation.

Reviewer #1: All comments have been addressed

2. Is the manuscript technically sound, and do the data support the conclusions?

Reviewer #1: Yes

3. Has the statistical analysis been performed appropriately and rigorously? 

Reviewer #1: Yes

4. Have the authors made all data underlying the findings in their manuscript fully available?

Reviewer #1: Yes

5. Is the manuscript presented in an intelligible fashion and written in standard English?

Reviewer #1: Yes

6. Review Comments to the Author

Reviewer #1: This revised manuscript has been clearly improved and all the points raised by this reviewer have been properly answered.

7. PLOS authors have the option to publish the peer review history of their article (what does this mean?). If published, this will include your full peer review and any attached files.

Reviewer #1: No

---

## [Editor Report · Acceptance letter]

1 Nov 2024

PONE-D-24-27298R1 

PLOS ONE

Dear Dr. Johnson, 

I'm pleased to inform you that your manuscript has been deemed suitable for publication in PLOS ONE. Congratulations! Your manuscript is now being handed over to our production team.

Kind regards, 

on behalf of

Prof. Vladimir Trajkovic 

Academic Editor

PLOS ONE